

# Sterile seed germination and seedling cultivation of *Idesia polycarpa*

Zhangtai Niu[1,2,3,4], Juan Xiao[1,2,3,4], Chuxi Hu[1,2,3,4],
Yunchen Yang[1,2,3,4], Sijing Shi[1,2,3,4], Xiaoyu Lu[1,2,3,4], Yian Yin[1,2,3,4],
Ze Li[1,2,3,4] and Lingli Wu[1,2,3,4]

[1] Lutou National Station for Scientific Observation and Research of Forest Ecosystem in Hunan Province, Changsha, Hunan, China
[2] State Key Laboratory of Utilization of Woody Oil Resource, Central South University of Forestry and Technology, Changsha, Hunan, China
[3] Key Laboratory of Cultivation and Protection for Non-wood Forest Trees, Ministry of Education, Central South University of Forestry and Technology, Changsha, Hunan, China
[4] Key Laboratory of Non-wood Forest Products, Forestry Ministry, Central South University of Forestry and Technology, Changsha, Hunan, China

Corresponding author
Lingli Wu, wulingli0307@163.com

## ABSTRACT

**Background:** *Idesia polycarpa* Maxim. is a high-quality, high-yield, edible oil tree species native to eastern Asia, where it plays important roles in ensuring national food and oil security, promoting ecological development, and facilitating rural revitalization. However, the commercial development of *I. polycarpa* has been hampered by the fact that it is primarily propagated by seeds, the required dormancy of which leads to low natural germination rates. Tissue culture technology offers the advantages of rapid propagation, high multiplication rates, and independence from seasonal factors, enabling the rapid production of large quantities of high-quality seedlings. The aim of this study was to establish an efficient aseptic germination system for *I. polycarpa* seeds.

**Methods:** This study utilized *I. polycarpa* seeds as the experimental material to investigate the effects of different disinfection times, basic medium variations, activated carbon (AC) concentrations, and the types and concentrations of plant growth regulators (PGRs) on aseptic germination. Subsequently, sterile seedlings were used as explants to screen for the effects of sucrose concentration and the types and concentrations of PGRs on rooting. The study also investigated how different substrate ratios and container types influenced the post-transplant survival rate of tissue-cultured *I. polycarpa* seedlings.

**Results:** The results showed that the optimal time was 10 min for *I. polycarpa* seed disinfection with 0.1% $HgCl_2$. The most suitable medium for *I. polycarpa* seed germination was 1/2 MS medium supplemented with $GA_3$ (1.0 mg·$L^{-1}$) and AC (1.0 g·$L^{-1}$), achieving a germination rate of 96.0%. A sucrose concentration of 10.0 g·$L^{-1}$ was most beneficial for rooting. When using a single plant growth regulator, indole-3-butyric acid (IBA) had the most significant effect on *I. polycarpa* root induction. The optimal medium for root development was Murashige and Skoog (MS) medium supplemented with IBA (0.3 mg·$L^{-1}$) and α-naphthyl acetic acid (NAA) (0.5 mg·$L^{-1}$), resulting in a 100% rooting rate and an average of 22.17 roots. These roots had an average length of 3.4 cm and were abundant and vigorous. Tissue-cultured seedlings were transplanted into transparent plastic cups containing a mixed substrate of organic nutrient soil (BALTIC PEAT), perlite, and vermiculite in

a ratio of 2:1:1 (V/V/V). They grew vigorously, with a survival rate as high as 96.67%. The findings of this study can provide technical support for the factory breeding of *I. polycarpa* seedlings.

# INTRODUCTION

*Idesia polycarpa* Maxim. (family Salicaceae) is a tall, perennial deciduous tree widely distributed along low-altitude hillsides and valleys across China (*Wang et al., 2023*), primarily in Sichuan, Guizhou, Shaanxi, Hubei, and Henan, but also in several other provinces. Abroad, it is mostly found in Japan, the Korean Peninsula, and the Russian Far East. The fruit of *I. polycarpa* contains up to 35% oil, with unsaturated fatty acids comprising 88.67%, of which linoleic acid accounts for >80%. *I. polycarpa* is therefore considered a "tree oil depot". The health benefits of *I. polycarpa* oil are due to its high contents of vitamin E, squalene, sterols, and other bioactive compounds known to be effective in preventing cardiovascular and cerebrovascular diseases (*Li et al., 2024*). Moreover, *I. polycarpa* oil is also widely utilized in biodiesel production, industrial oils, pharmaceuticals, and cosmetics (*Xiang et al., 2023*). Other features of interest include its suitability for ecological conservation, furniture production, landscaping, and ornamental use (*Sui, 2011*). As a multipurpose, multifunctional oil tree species, *I. polycarpa* has been included in China's National Strategic Reserve Forest list since April 2020 (*Wu et al., 2023*).

The industries that rely on *I. polycarpa* have grown rapidly in recent years, and the demand for seedlings has increased accordingly. However, commercial development of the tree has been hindered by two major constraints (*Liu et al., 2009*). First, *I. polycarpa* seedlings mainly originate from wild sources, such that their traits are unstable and their yields uncertain (*Liu et al., 2022*). Second, the propagation of *I. polycarpa* through sowing is hampered by the required dormancy period of the seeds and thus their low germination rates under natural conditions.

Plant tissue culture has emerged as an important methodology for both large-scale seedling propagation and the development of superior cultivars (*Zhou, Deng & Tang, 2020*). Aseptic seeding and tissue culture experiments may enable the expansion of *I. polycarpa* populations and the large-scale cultivation of high-quality seedlings. Specifically, the propagation of *I. polycarpa* via aseptic seeding can improve seed germination rates and shorten the seedling cultivation cycle, thereby overcoming the low germination rates and seasonal restrictions associated with artificial seeding. However, the aseptic germination of *I. polycarpa* seeds has been examined in only a few studies. *Liu (2009)* and *Shen et al. (2015)* determined the survival rate of disinfected *I. polycarpa* seeds but did not describe the culture conditions in detail. *Li (2023)* established an aseptic

germination system for *I. polycarpa* seeds, but the induction rate of the sterile seedlings was only 70%.

The main step in tissue culture is rooting, and the root induction aims to produce seedlings with numerous thick roots and robust branches, as these traits are key to seedling survival following transplantation. An appropriate medium and the application of auxins and other plant growth regulators (PGRs) can promote root development, such as indole-3-acetic acid (IAA), indole-3-butyric acid (IBA), α-naphthyl acetic acid (NAA), and 6-benzyladenine (6-BA). *Jiang et al. (2006)* utilized 1/2 Murashige and Skoog (MS) medium as the basal medium and added IBA and NAA to facilitate *I. polycarpa* root induction, achieving a rooting rate of 87%. *Chen et al. (2022)* used a combination of IAA, IBA, and NAA to induce rooting in tissue culture seedlings of the elite variety "Exuan 1", attaining rooting rates of up to 98%. The optimal medium for adventitious bud induction and rooting was determined to be 1/2 MS + 0.5 mg·L$^{-1}$ IBA + 0.1 mg·L$^{-1}$ IAA + 20 g·L$^{-1}$ sucrose, achieving a rooting efficiency of 83% (*Wang, 2024*). *Shen (2015)* conducted shade treatments and tested various substrate ratios for the transplantation of *I. polycarpa* tissue culture seedlings, ultimately achieving a survival rate of 65.85%. The transplantation of tissue culture seedlings of *I. polycarpa* using a mixture of vermiculite, perlite, and peat in a volumetric ratio of 6:3:3 resulted in seedling survival rates of up to 95.5% (*Hong et al., 2023*). Despite encouraging results, these studies provided little information on cultivation conditions, the selection of seedling containers, and transplantation techniques. Therefore, there remains a need for in-depth, systematic research on efficient and simple acclimation techniques for *I. polycarpa* tissue culture seedlings, as well as the establishment of a comprehensive acclimation system for tissue culture of this species.

This study utilizes mature *I. polycarpa* seeds as raw material to investigate various factors affecting seed germination, identify the optimal culture medium for root induction, and determine the best conditions for seedling hardening and transplantation. The study aims to develop an efficient and rapid aseptic germination system for *I. polycarpa* seeds and establish a comprehensive set of seedling cultivation protocols, providing technical support and practical evidence for the large-scale, factory-based production of *I. polycarpa* seedlings in the future.

## MATERIALS AND METHODS

### Plant material

Mature *I. polycarpa* seeds were harvested from Qishe Town, Xingyi, Guizhou Province, China (24°56′N, 104°47′E) in late November 2022. Following pericarp removal, the seeds were thoroughly washed to remove residual debris. Damaged seeds, floating impurities, and shriveled or empty seeds were excluded. The washing procedure was repeated 2–3 times. The seeds were air-dried indoors and stored at temperatures ranging from 0 °C to 10 °C until use.

### Reagents

Indole-3-acetic acid (IAA; Solarbio, Beijing, China); α-Naphthaleneacetic acid (NAA; Solarbio, Beijing, China); gibberellic acid (GA$_3$; Solarbio, Beijing, China);

6-benzylaminopurine (6-BA; Solarbio, Beijing, China); indole-3-butyric acid (IBA; Solarbio, Beijing, China). MS medium (Solarbio, Beijing, China); 1/2 MS medium (Solarbio, Beijing, China); woody plant medium (WPM, BKMAM, Changde, China); agar powder (Solarbio, Beijing, China); sucrose (analytical reagent grade; Guoyao, Shanghai, China); anhydrous ethanol (Hengxing, Tianjin, China); mercuric chloride (0.1% w/v), Rooting powder (10.0–15.0 g·L$^{-1}$), activated charcoal (AC; Guoyao, Shanghai, China).

## Pretreatment of experimental materials

Full, healthy *I. polycarpa* seeds (1,000-seed weight: 4.007 g) that had been reserved for 1 month were subjected to pretreatment. Initially, the seeds were immersed in a 30% washing powder solution for 2 h, wrapped in gauze, and kneaded repeatedly to remove the waxy layer. The seeds were then rinsed with running water for 30 min. The cleaned seeds were soaked in pure water at 30–40 °C for 30 min and then placed in an ultra-clean worktable for further processing.

## Effects of different disinfection time on *I. polycarpa* seed germination

Seeds were surface-sterilized with a 0.1% HgCl$_2$ solution for 6, 8, 10, or 12 min. After disinfection, the seeds were rinsed 3–5 times with sterile water and then placed on sterile filter paper to absorb surface water before being inoculated onto the seed germination media. Each seed was cultivated in a glass culture bottle containing 50 mL of media. Ten seeds were inoculated per bottle, with 10 bottles per treatment and three replicates. After 30 days, the germination rate and contamination rate were calculated. Germination rate (%) = (30 d germinated seeds/inoculated seeds) × 100, contamination rate (%) = (30 d contaminated seeds/inoculated seeds) × 100.

## Effects of different basic media on *I. polycarpa* seed germination

The sterilized seeds were inoculated onto three different basic media, MS, 1/2 MS, and WPM, each supplemented with 1.0 mg·L$^{-1}$ GA$_3$ + 30.0 g·L$^{-1}$ sucrose + 8.0 g·L$^{-1}$ agar + 1.0 g·L$^{-1}$ AC. Ten seeds were inoculated per bottle, with 10 bottles per treatment and three replicates. After 30 days, the germination rate, contamination rate, and seedling height were measured.

## Effects of different concentrations of activated carbon on *I. polycarpa* seed germination

The sterilized seeds were cultured in 1/2 MS + 1.0 mg·L$^{-1}$ GA$_3$ + 30.0 g·L$^{-1}$ sucrose + 8.0 g·L$^{-1}$ agar, and varying concentrations of activated carbon (0.0, 1.0, 2.0, and 3.0 g·L$^{-1}$). The activated carbon was thoroughly stirred until no visible particles remained, and the most effective concentration for *I. polycarpa* seed germination was screened. Ten seeds were inoculated per bottle, with 10 bottles per treatment and three replicates. After 30 days, the germination rate, contamination rate, and seedling height were measured.

## Effects of different PGRs on *I. polycarpa* seed germination

The sterilized seeds were inoculated onto basic media containing varying concentrations of different PGRs (GA$_3$: 0.5, 1.0, 1.5, 2.0 mg·L$^{-1}$; 6-BA: 0.5, 1.0, 1.5, 2.0 mg·L$^{-1}$; NAA: 0.5, 1.0,

1.5, 2.0 mg·L$^{-1}$). Ten seeds were inoculated per bottle, with 10 bottles per treatment and three replicates. No PGR was added to the control group. After 30 days, the germination rate, contamination rate, and seedling height were measured to determine the most effective hormonal conditions for seed germination.

## Effects of different sucrose concentrations on rooting in *I. polycarpa* sterile seedlings

The sterile seedlings of *I. polycarpa* with heights of 4–7 cm were selected after 30 days of growth. The base of the hypocotyl was trimmed, and the seedlings were inoculated onto MS basal medium containing sucrose at a concentration of 10.0, 20.0, or 30.0 g·L$^{-1}$. Twenty bottles were inoculated per treatment. The rooting rate, average root length, and average number of roots were assessed 30 days after inoculation using the following formulas: Rooting rate (%) = (Number of rooting explants/Total explants) × 100, Average root length = Total root length/Number of roots, and Average root number = Total number of roots/Number of rooting plants.

## Effects of different PGRs on rooting in *I. polycarpa* sterile seedlings

The sterile seedlings of *I. polycarpa*, with heights between 4 and 7 cm after 30 days of growth, were selected, and the base of the hypocotyl was trimmed. Different concentrations of IBA (0.1, 0.3, 0.5, 0.7, 0.9 mg·L$^{-1}$) or IAA (0.05, 0.1, 0.5, 1.0, 1.5 mg·L$^{-1}$) or NAA (0.01, 0.05, 0.1, 0.5, 1.0 mg·L$^{-1}$) were added to MS basal medium. The rooting and growth of the seedlings were observed after 30 days, with the rooting rate, average root length, and average root number measured. Combinations of IBA (0.1, 0.3, 0.5 mg·L$^{-1}$) and NAA (0.1, 0.3, 0.5 mg·L$^{-1}$) were also tested to find the best combination for rooting. No PGR was added to the control group. One or two seedlings were inoculated per bottle, with10 bottles per treatment and three replicates.

## Effects of different substrate ratios on transplant survival rates in *I. polycarpa* sterile seedlings

The well-grown tissue culture seedlings were selected after 30 days of rooting and transplanted into transparent plastic cups filled with different substrate mixtures. The transparent plastic cups used for transplantation had a bottom diameter of 8 cm, height of 14.5 cm, and top diameter of 11.5 cm. Each cup was covered with a lid. An air vent (diameter: 3–6 mm) was made both at the side, 0.5 cm from the bottom, and in the lid to facilitate aeration. Three different substrate ratios were tested using organic nutrient soil, perlite, and vermiculite at volume ratios of 1:1:1, 2:1:1, and 1:2:1. The substrate water content was maintained above 70%. Thirty seedlings were transplanted per treatment, and the growth was observed with related indexes counted after 25 days. The transplanting survival rate (%) = (number of surviving seedlings/total number of transplanted seedlings) × 100.

Prior to transplantation, the seedlings were carefully washed with water to remove residual culture medium and then dipped in 10.0–15.0 g·L$^{-1}$ of rooting powder solution for 5–15 s. The seedlings were cultured in a light chamber for 10 days and then transferred to a

greenhouse, where they were cultivated for a further 10–15 days. The greenhouse conditions were controlled to a temperature of 25–35 °C and relative humidity of 60–80%. Subsequently, the seedlings were transplanted into seedling bags and continued to culture for 7 days in the greenhouse. The nursery was shaded with a sunshade net, and substrate moisture was monitored and adjusted as needed. A 0.1% urea solution and a fungicide were applied monthly.

## Effects of different transplant container types on transplant survival rates in *I. polycarpa* tissue culture seedlings

To evaluate the effect of the type of transplant container on seedling survival, well-grown tissue culture seedlings were selected after 30 days of rooting and transplanted into either transparent plastic cups or seedling trays. In both cases, the substrate consisted of organic nutrient soil, perlite, and vermiculite at a 2:1:1 volume ratio. The transplantation procedure used for the cups was the same as that described above. For seedlings in the seedling trays, the procedure was as follows: seedlings were kept in closed tissue culture bottles for 4 days and then exposed to air in the greenhouse for 3 days. Then, they were rinsed to remove the culture medium, dipped into 10.0–15.0 g·L$^{-1}$ rooting powder solution for 5–15 s, and directly transplanted into 32-hole seedling trays. Transplanted seedlings in the cups and trays were maintained under similar conditions in the greenhouse (20–35 °C, 60–80% relative humidity). After 25 days, the transplantation survival rate, average seedling height, and overall growth were recorded. Data were collected on seedling growth parameters, and a comparative analysis was performed to identify the most effective transplanting container for improving the survival rate and growth of *I. polycarpa* tissue culture seedlings.

### Culture conditions

MS or 1/2 MS was mainly used as the basic medium, with sucrose content of 3% (rooting of 1%), Agar content of 0.8% (rooting of 0.7%), pH 5.7~5.9, and autoclaving conditions of 116 °C 30 min (MS and 1/2 MS medium) or 121 °C 15 min (WPM medium). PGRs were added to the culture medium at different stages to support specific growth needs. The temperature of the culture room was 25 ± 2 °C, the light intensity was 50–60 µmol·m$^{-2}$·s$^{-1}$, and the light time was 12 h·d$^{-1}$.

### Statistical analysis

All data presented in the tables were sourced from the raw data provided in the Supplemental Files. Data collection and statistical analyses were performed using Microsoft Excel (2016) and IBM SPSS Statistics for Windows, version 26 (IBM Corp., Armonk, NY, USA). The results were analyzed by one-way ANOVA. The data are presented as the mean ± standard deviation (SD). Duncan's multiple range test was used to assess significant differences among means at *P < 0.05*. Image manipulation using Photoshop 2022.

**Table 1 Effects of the duration of disinfection with 0.1% HgCl₂ on *I. polycarpa* seed germination and contamination rates.**

| 0.1% HgCl₂/min | Germination rate/% | Pollution rate/% |
|---|---|---|
| 6 | 77.00 ± 2.64[c] | 15.67 ± 2.40[a] |
| 8 | 84.33 ± 0.88[b] | 9.00 ± 1.00[b] |
| 10 | 96.00 ± 1.15[a] | 0.33 ± 0.33[c] |
| 12 | 79.67 ± 1.20[bc] | 0.00 ± 0.00[c] |

Note:
Data are means ± standard error (SE). Different letters indicate significant differences ($P < 0.05$; one-way analysis of variance (ANOVA), followed by least significant difference (LSD) test).

## RESULTS

### Effects of disinfection time on *I. polycarpa* seed germination

The contamination and germination rates of *I. polycarpa* seeds subjected to the different disinfection treatments for 30 days are summarized in Table 1. Seeds treated for 10 min had a significantly higher germination rate ($96.00 ± 1.15\%$; $P < 0.05$) than those in the other treatments. Seed germination began 8 days post-inoculation, as observed by radicle emergence. Most cotyledons were fully expanded by day 15, at which time the outer seed coat fell off. By day 30, the sterile seedlings had grown robustly. A disinfection time of 6 min using 0.1% HgCl₂ resulted in the highest contamination rate (15.67%). As the disinfection time increased, the contamination rate decreased; however, the germination rate decreased after an initial increase, indicating detrimental effects of prolonged disinfection on seed viability. Based on the germination and contamination rates, a disinfection time of 10 min using 0.1% HgCl₂ was considered optimal for *I. polycarpa* seed sterilization.

### Effects of basal medium on *I. polycarpa* seed germination

There were significant differences in the germination rates of *I. polycarpa* seeds across various basal media (Table 2). After 30 days, seeds inoculated on 1/2 MS medium exhibited a germination rate of 94.67%, and the average seedling height reached 4.20 cm. These seedlings had emerald green leaves and showed rapid growth during the later stages. In contrast, seeds inoculated on WPM and MS media germinated slowly and irregularly. Although most seeds eventually germinated after 30 days, some exhibited deformities that hindered normal growth. Thus, 1/2 MS medium was identified as the most suitable for aseptic germination of *I. polycarpa* seeds.

### Effects of activated carbon concentration on *I. polycarpa* seed germination

Different activated carbon significantly influenced the germination of *I. polycarpa* seeds (Table 3). Seeds cultured with activated carbon consistently exhibited higher germination rates compared to the control group without supplementation. When the activated carbon content was 1.0 g·L⁻¹, the highest germination rate was 94.67%. After 30 days, the plant height was up to 4.13 cm, and the hypocotyl and leaves were dark green, with developed roots and robust plants. However, as the concentration of activated carbon increased, the

**Table 2 Effects of different media on seed germination rates and seedling height in *I. polycarpa*.**

| Types of medium | Germination rate/% | Seedling height/cm |
|---|---|---|
| 1/2MS | 94.67 ± 1.20[a] | 4.20 ± 0.21[a] |
| MS | 88.33 ± 1.45[b] | 3.13 ± 0.15[b] |
| WPM | 82.33 ± 1.20[c] | 2.50 ± 0.21[c] |

Note:
Data are means ± SE. Different letters indicate significant differences ($P < 0.05$; one-way ANOVA, followed by LSD test).

**Table 3 Effects of different activated carbon (AC) concentrations on seed germination and seedling height in *I. polycarpa*.**

| Activated carbon concentrations/g·L$^{-1}$ | Germination rate/% | Seedling height/cm |
|---|---|---|
| 0 | 72.00 ± 1.52[c] | 2.01 ± 0.12[c] |
| 1.0 | 94.67 ± 1.20[a] | 4.13 ± 0.15[a] |
| 2.0 | 79.00 ± 1.73[b] | 3.50 ± 0.10[b] |
| 3.0 | 73.33 ± 0.67[c] | 2.13 ± 0.11[c] |

Note:
Data are means ± SE. Different letters indicate significant differences ($P < 0.05$; one-way ANOVA, followed by LSD test).

seed germination rate decreased, and seedling growth slowed. At a concentration of 3.0 g·L$^{-1}$, many seeds germinated but only produced green bud points without further growth into full seedlings. The sterile seedlings without added activated carbon exhibited slow growth in the later stage, featuring light green hypocotyls, thin and weak plants, and slightly yellow leaves.

## Effects of different PGRs and PGR concentrations on *I. polycarpa* seed germination

The addition of PGRs improved the germination of *I. polycarpa* seeds to varying degrees (Table 4). The germination rate of control non-treated seeds was 73.33%. The addition of GA$_3$, NAA, and 6-BA to the culture medium resulted in a concentration-dependent increase in the germination rate. The optimal concentrations of GA$_3$ and NAA were 1.0 and 0.5 mg·L$^{-1}$, which yielded germination rates of 96.00% and 88.00%, respectively, with seedling growth increased by 30.9% and 20.0% compared to the control group. 6-BA had the poorest overall effect on germination, delaying the start of germination, emergence through the seed coat, and seedling growth. At the optimal concentration of GA$_3$, the seed coat cracked by the 6th day, the radicle protruded by the 8th day, and the cotyledons unfolded with tender green leaves by the 14th day, exhibiting rapid growth and well-developed adventitious roots. In conclusion, the best PGR for inducing germination of *I. polycarpa* sterile seedlings was GA$_3$, and the most suitable culture medium for seed germination was 1/2 MS + 1.0 mg·L$^{-1}$ GA$_3$ + 1.0 g·L$^{-1}$ AC (Fig. 1A).

## Effects of different sucrose concentrations on rooting in *I. polycarpa* sterile seedlings

Following 6 days of cultivation, protrusions emerged at the stem base segments of vigorous sterile seedlings and gradually differentiated into root primordia. After 30 days, the roots

**Table 4 Effects of different hormones and their concentrations on seed germination and seedling height in *I. polycarpa*.**

| Types of hormone | Concentration/mg·L$^{-1}$ | Germination rate/% | Seedling height/cm |
|---|---|---|---|
| 0 | 0 | 73.33 ± 1.86[e] | 2.93 ± 0.19[cd] |
| GA3 | 0.5 | 89.00 ± 1.15[b] | 3.23 ± 0.14[bcd] |
| | 1.0 | 96.00 ± 0.58[a] | 4.13 ± 0.08[a] |
| | 1.5 | 88.67 ± 0.67[b] | 3.63 ± 0.03[b] |
| | 2.0 | 80.67 ± 1.20[cd] | 3.07 ± 0.08[cd] |
| NAA | 0.5 | 88.00 ± 2.08[b] | 3.27 ± 0.12[bc] |
| | 1.0 | 82.67 ± 1.45[c] | 3.17 ± 0.13[cd] |
| | 1.5 | 80.33 ± 1.45[cd] | 2.93 ± 0.08[cd] |
| | 2.0 | 77.67 ± 1.86[cde] | 2.80 ± 0.05[d] |
| 6-BA | 0.5 | 76.33 ± 1.86[de] | 2.80 ± 0.05[d] |
| | 1.0 | 82.33 ± 1.86[c] | 3.20 ± 0.21[cd] |
| | 1.5 | 76.00 ± 2.00[de] | 3.23 ± 0.08[bcd] |
| | 2.0 | 74.00 ± 1.52[e] | 3.00 ± 0.25[cd] |

Note:
Data are means ± SE. Different letters indicate significant differences ($P < 0.05$; one-way ANOVA, followed by LSD test).

had elongated and were well developed (Fig. 1B). As shown in Table 5, the best effect on root development in *I. polycarpa* was obtained using treatment 2, which resulted in an average root length of 2.66 cm and average root number of 16.85. The poorest effect was obtained with treatment 3, with an average root length of 2.38 cm and average root number of 12.30. Although average root length was not significantly affected by the sucrose concentration, the number of roots gradually decreased as the concentration increased, such that plant growth was weakened and occasional leaf yellowing was observed. Based on these observations, a sucrose concentration of 10.0 g·L$^{-1}$ was determined to be optimal for rooting.

**Effects of different auxins on rooting in *I. polycarpa* sterile seedlings**

As shown in Table 6, the three auxins affected the rooting rates of *I. polycarpa* tissue culture seedlings, with significant differences between the control group and the experimental group. In the control group, the rooting rate of sterile seedlings was only 33.33% and the average root length was 0.98 cm, while under auxin supplementation, the best performance was obtained with medium containing 0.7 mg·L$^{-1}$ IBA, which resulted in a rooting rate of 96.77%, root length of 3.05 cm, and the highest average root number (13.60). However, the rooting rate initially increased and then decreased in response to increasing IBA concentrations. At higher IBA concentrations, a callus formed at the stem base, and the roots became thicker and swollen. In seedlings exposed to IAA at a concentration of 1.0 mg·L$^{-1}$, the rooting rate, average root length, and average root number were highest, at 92.00%, 2.38 cm, and 10.37, respectively. Overall, the worst performance was that of medium containing 0.01 mg L$^{-1}$ NAA, which yielded a rooting rate of 72.00%, average root length of 1.65 cm, and average root number of 6.87. By contrast, among the NAA treatments, a concentration of 0.05 mg·L$^{-1}$ resulted in the

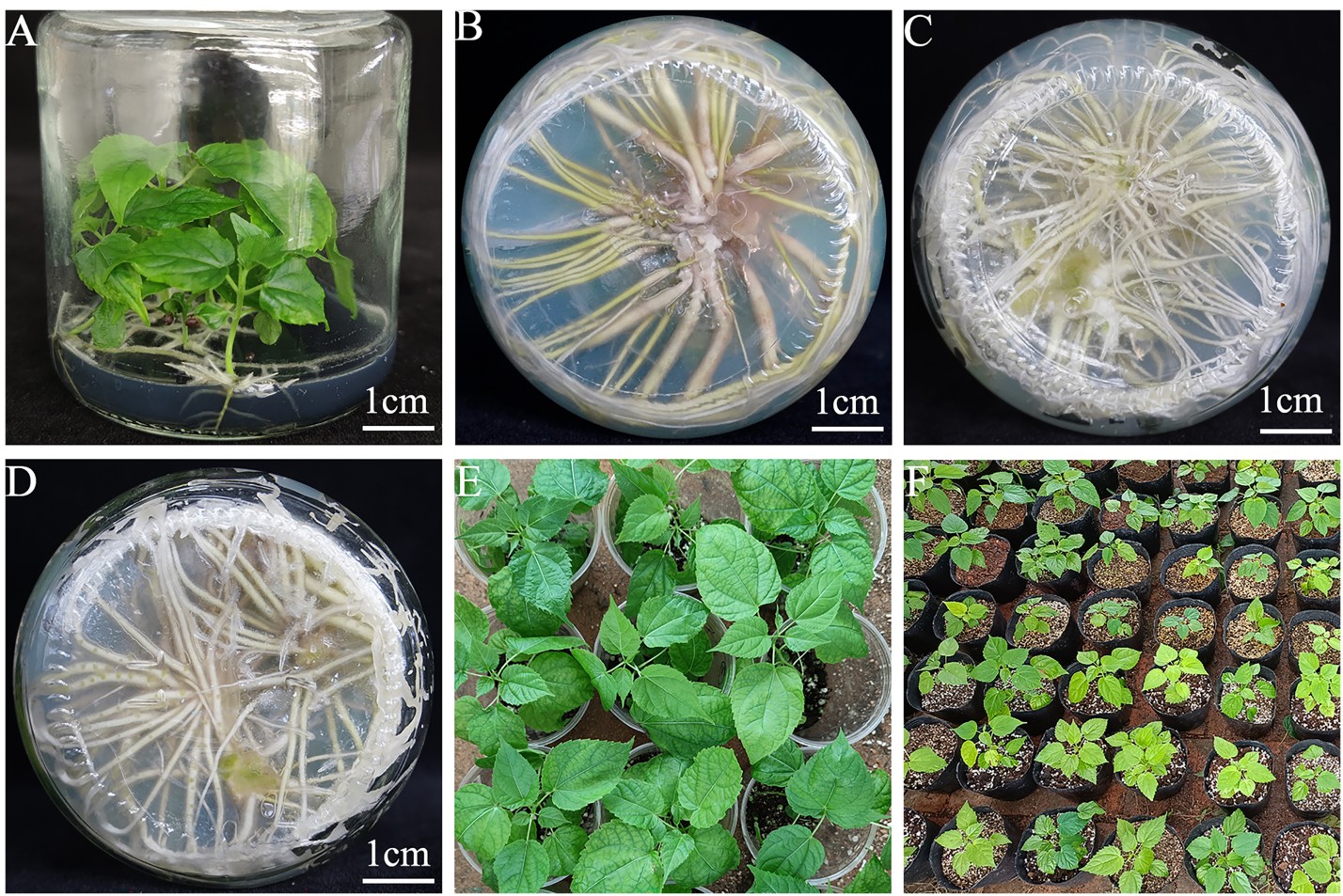

**Figure 1 Sterile germination and rapid propagation of *I. polycarpa*.** (A) Seed germ-free germination for 30 days. (B) Sucrose concentration of 10 g·L$^{-1}$ rooting for 30 days. (C) IBA 0.7 mg·L$^{-1}$ rooting for 30 days. (D) 0.3 mg·L$^{-1}$ IBA + 0.5 mg·L$^{-1}$ NAA rooting for 30 days. (E) Transplanting to transparent plastic cup for 30 days growth. (F) The overall growth of the seedling bags.

**Table 5 Effects of different sucrose concentrations on *I. polycarpa* roots.**

| Treatment | Sucrose concentrations/g·L$^{-1}$ | Rooting rate/% | Average root length/cm | Average number of roots/roots |
|---|---|---|---|---|
| 1 | 10 | 95.00 | 2.66 ± 0.21[a] | 16.85 ± 1.50[a] |
| 2 | 20 | 95.00 | 2.50 ± 0.28[a] | 14.60 ± 1.71[ab] |
| 3 | 30 | 85.00 | 2.38 ± 0.15[a] | 12.30 ± 0.95[b] |

**Note:**
Data are means ± SE. Different letters indicate significant differences ($P < 0.05$; one-way ANOVA, followed by LSD test).

highest rooting rate (94.00%), average root length (2.22 cm), and average root number (12.80). At higher NAA concentrations, the proliferating callus tissue negatively impacted root development such that the root system was clumped and the leaves exhibited yellowing or even abscission, which hindered the subsequent acclimatization and transplantation of the seedlings. Overall, 0.7 mg·L$^{-1}$ was identified as the optimal treatment for promoting adventitious root development in *I. polycarpa* sterile seedlings (Fig. 1C).

**Table 6 Effects of different auxins and their concentrations on *I. polycarpa* roots.**

| Auxins concentrations/mg·L$^{-1}$ | | | Rooting rate/% | Average root length/cm | Average number of roots/roots |
|---|---|---|---|---|---|
| IBA | IAA | NAA | | | |
| 0 | 0 | 0 | 33.33 ± 3.33[h] | 0.98 ± 0.14[h] | 1.17 ± 0.03[f] |
| 0.1 | – | – | 87.33 ± 0.88[de] | 1.99 ± 0.02[def] | 8.13 ± 0.98[de] |
| 0.3 | – | – | 87.67 ± 0.67[cde] | 2.32 ± 0.11[bc] | 8.83 ± 0.54[d] |
| 0.5 | – | – | 95.00 ± 1.15[a] | 2.49 ± 0.06[b] | 11.57 ± 0.30[bc] |
| 0.7 | – | – | 96.77 ± 0.67[a] | 3.05 ± 0.04[a] | 13.60 ± 0.66[a] |
| 0.9 | – | – | 93.33 ± 1.20[ab] | 2.27 ± 0.25[bcd] | 13.43 ± 0.28[a] |
| – | 0.05 | – | 81.33 ± 0.88[f] | 1.90 ± 0.05[efg] | 6.83 ± 0.32[e] |
| – | 0.1 | – | 85.00 ± 2.88[def] | 1.94 ± 0.03[efg] | 7.23 ± 0.58[e] |
| – | 0.5 | – | 89.33 ± 1.20[bcd] | 2.09 ± 0.06[cdef] | 8.20 ± 0.15[de] |
| – | 1.0 | – | 92.00 ± 0.58[abc] | 2.38 ± 0.15[bc] | 10.37 ± 0.12[c] |
| – | 1.5 | – | 83.00 ± 1.15[ef] | 2.11 ± 0.02[cdef] | 7.77 ± 0.38[de] |
| – | – | 0.01 | 72.00 ± 1.15[g] | 1.65 ± 0.03[g] | 6.87 ± 0.29[e] |
| – | – | 0.05 | 94.00 ± 1.00[a] | 2.22 ± 0.05[bcde] | 12.80 ± 0.35[ab] |
| – | – | 0.1 | 84.67 ± 0.88[def] | 2.09 ± 0.08[cdef] | 11.47 ± 0.15[bc] |
| – | – | 0.5 | 83.33 ± 1.45[ef] | 1.97 ± 0.09[def] | 10.97 ± 0.47[c] |
| – | – | 1.0 | 82.33 ± 1.45[f] | 1.83 ± 0.07[fg] | 10.17 ± 0.59[c] |

Note:
Data are means ± SE. Different letters indicate significant differences ($P < 0.05$; one-way ANOVA, followed by LSD test).

**Table 7 Effect of combined IBA and NAA concentrations on rooting performance in *I. polycarpa*.**

| Treatment | Auxin concentration/mg·L$^{-1}$ | | Rooting rate/% | Root length/cm | Average number of roots/roots |
|---|---|---|---|---|---|
| | IBA | NAA | | | |
| T1 | 0.1 | 0.1 | 90.33 ± 0.67[c] | 1.96 ± 0.08[d] | 9.93 ± 0.89[e] |
| T2 | 0.1 | 0.3 | 96.67 ± 0.33[b] | 2.22 ± 0.14[cd] | 14.83 ± 0.58[bcd] |
| T3 | 0.1 | 0.5 | 72.33 ± 0.88[e] | 2.48 ± 0.05[bc] | 17.07 ± 0.85[bc] |
| T4 | 0.3 | 0.1 | 83.33 ± 0.88[d] | 2.26 ± 0.14[cd] | 12.93 ± 1.49[bc] |
| T5 | 0.3 | 0.3 | 89.67 ± 0.33[c] | 2.84 ± 0.19[b] | 17.00 ± 1.47[de] |
| T6 | 0.3 | 0.5 | 100.00 ± 0.00[a] | 3.40 ± 0.20[a] | 22.17 ± 2.04[a] |
| T7 | 0.5 | 0.1 | 95.33 ± 0.67[b] | 2.49 ± 0.12[bc] | 18.33 ± 0.73[b] |
| T8 | 0.5 | 0.3 | 91.00 ± 0.58[c] | 2.42 ± 0.04[c] | 16.00 ± 1.21[bcd] |
| T9 | 0.5 | 0.5 | 90.67 ± 0.33[c] | 2.25 ± 0.08[cd] | 14.37 ± 0.55[cd] |

Note:
Data are means ± SE. Different letters indicate significant differences ($P < 0.05$; one-way ANOVA, followed by LSD test).

Vigorous sterile seedlings (4–7 cm in height) were inoculated onto MS medium supplemented with different combinations of IBA and NAA (Table 7). After 8 days, adventitious roots began to form; after 30 days, the roots were well developed (Fig. 1D), and the seedlings were lush, green, and showed good growth. Different auxin combinations induced root formation to different degrees. Compared with the respective single treatments, the combination of IBA and NAA significantly enhanced both the rooting rate and root number. The most effective combination promoting adventitious

**Table 8 Effects of different substrate ratios on _I. polycarpa_ seedling growth.**

| Treatment | Substrate ratio/V:V:V | Transplanting survival rate/% | Growth status (25 d) |
|---|---|---|---|
| A1 | Organic soil: perlite: vermiculite = 1:1:1 | 90.00 ± 2.89[ab] | Plants grow well; leaves slightly yellow |
| A2 | Organic soil: perlite: vermiculite = 2:1:1 | 96.67 ± 1.67[a] | Plants grow well, robust, and fast; leaves are light green |
| A3 | Organic soil: perlite: vermiculite = 1:2:1 | 86.67 ± 1.67[b] | Plants grow slowly; leaves are green |

Note:
  Data are means ± SE. Different letters indicate significant differences ($P < 0.05$; one-way ANOVA, followed by LSD test).

root growth in _I. polycarpa_ sterile seedlings was treatment 6, which consisted of 0.3 mg·L$^{-1}$ IBA and 0.5 mg·L$^{-1}$ NAA. Under these conditions, the rooting rate was 100%, the average root length was 3.40 cm, and the average root number was 22.17. Moreover, treatment 6 was significantly more effective than the other treatments in producing thick roots with few lateral roots ($P < 0.05$). Treatment 1 also showed significant differences from the other treatments, with a dramatically shorter average root length (1.96 cm) and lower average root number (9.93). These results indicated that the combination of 0.3 mg·L$^{-1}$ IBA and 0.5 mg·L$^{-1}$ NAA (T6) was the most suitable for promoting adventitious root growth in _I. polycarpa_ sterile seedlings.

### Effects of different substrate ratios on transplant survival rates in _I. polycarpa_ tissue culture seedlings

Statistical analysis of Table 8 indicated that the mixed substrate containing organic nutrient soil, perlite, and vermiculite in a 2:1:1 volumetric ratio achieved the highest transplant survival rate (96.67%) with optimal seedling growth (Fig. 1E). The $A_2$ treatment likely provided excellent water retention and air permeability, thereby promoting optimal root system development. The lowest survival rate (86.67%) was obtained with treatment $A_3$, which reduced the amount of root contact due to the increased porosity resulting from the high perlite content, reducing root development. These results demonstrate that treatment $A_2$ can be considered effective for producing tissue-cultured _I. polycarpa_ seedlings. Subsequently, the seedlings were transplanted into nursery bags containing the same substrate ratios and further cultivated in the greenhouse for 7 days. During rainy days and in the mornings and evenings, they were relocated to a shaded nursery covered with sunshade nets (Fig. 1F).

### Effects of different transplantation container types on _I. polycarpa_ tissue culture seedlings

The growth performance of _I. polycarpa_ tissue-cultured seedlings in transparent plastic cups _versus_ seedling trays is presented in Table 9. Seedlings in the seedling trays showed signs of stress within the first 3 days of transplantation, indicating their poor adaptation to the new environment, and many subsequently died. However, after approximately 10 days, there were no significant differences in height, ground diameter, or leaf count between surviving seedlings in trays and those in transparent plastic cups. Seedlings grown in cups thrived, with a survival rate of 96.67%, average height of 20.35 cm, ground diameter of 2.95 mm, average of 9.30 leaves/seedling, and excellent growth vigor after 25 days (Fig. 2).

**Table 9 Effects of different container types on *I. polycarpa* seedling growth.**

| Type of container | Day of transplantation/d | Transplanting survival rate/% | Seedling height/cm | Ground diameter/mm | Number of leaf | Growth status |
|---|---|---|---|---|---|---|
| Transparent plastic cups | 10 | – | 10.57 ± 0.57[d] | 1.07 ± 0.05[d] | 6.60 ± 0.37[cd] | Plant grew rapidly from 20 to 30 days; leaves were large and bright green; roots were developed |
| | 20 | – | 15.90 ± 0.80[b] | 2.22 ± 0.13[b] | 8.00 ± 0.52[abc] | |
| | 25 | 96.67 | 20.35 ± 0.98[a] | 2.95 ± 0.15[a] | 9.30 ± 0.56[a] | |
| Growing hole trays | 10 | – | 8.47 ± 0.53[d] | 1.05 ± 0.05[d] | 6.30 ± 0.47[d] | Plants grew slowly; some leaves appeared yellowish and fell off; growth is weak |
| | 20 | – | 10.68 ± 0.72[d] | 1.48 ± 0.06[d] | 7.10 ± 0.43[bcd] | |
| | 25 | 73.33 | 12.97 ± 0.89[c] | 1.87 ± 0.08[c] | 8.20 ± 0.49[ab] | |

**Note:**
Data are means ± SE. Different letters indicate significant differences ($P < 0.05$; one-way ANOVA, followed by LSD test).

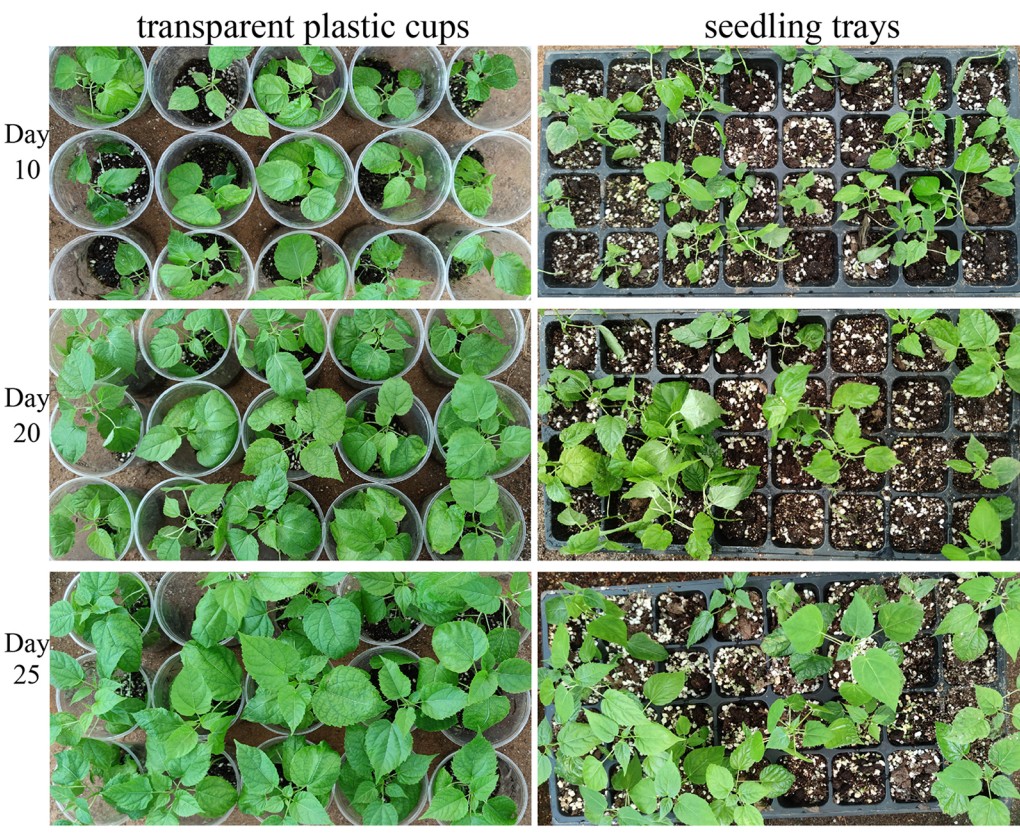

**Figure 2 Effects of different transplant containers on tissue culture plantlets of *I. polycarpa*.**

Seedlings in trays grew more slowly; their stems were thin and their leaves eventually turned yellow and abscised. These results demonstrated that transparent plastic cups provided a more favorable environment for the growth of *I. polycarpa* tissue culture seedlings, including by promoting root development.

## DISCUSSION

### Factors improving *I. polycarpa* seed germination

In plant tissue culture, seed sterilization is required to control contamination and thus allow the production of sterile seedlings. In this study, *I. polycarpa* seeds were initially disinfected with 70% ethanol for 2 min, followed by 0.1% $HgCl_2$ for 5 min, which reduced the contamination rate to 2%. In a previous study, when this treatment was applied in combination with a 30% hydrogen peroxide solution, the seed survival rate was 90% (*Shen, 2015*). However, our results showed that *I. polycarpa* seeds can be sterilized by simply treating them with 0.1% $HgCl_2$ for 10 min; the survival rate with this method was 96.00%. The study demonstrated that mature *I. polycarpa* seeds with intact dense testa maintained viable germination capacity, suggesting that mechanical restriction is not the predominant factor of dormancy. The germination rate significantly increased from 73.33% (control group) to 96.00% ($P < 0.05$) when combining 1.0 mg·$L^{-1}$ $GA_3$ supplementation with 0–10° C stratification, revealing synergistic interaction between gibberellin signaling and low-temperature regulation in physiological dormancy alleviation in *I. polycarpa* seeds. These findings are consistent with those reported by *Sun (2019)*. Seed germination was most effectively promoted by treating the seeds with 1.0 mg·$L^{-1}$ $GA_3$, followed by 0.05 mg·$L^{-1}$ NAA, with 6-BA having the weakest effect. These findings are inconsistent with previous research on *Streptocaulon griffithii*, in which 6-BA was found to be more effective in inducing the development of sterile seedlings (*Xiao et al., 2023*).

A suitable basic culture medium can facilitate the rapid germination of explants and enhance their growth. For example, the addition of AC provides a dark environment for seed germination, improves gas conditions in the culture medium, and adsorbs toxic substances such as the quinones and phenols produced during plant growth (*Zhao et al., 2017*). In our experiment, 1.0 g·$L^{-1}$ AC was therefore included to promote *I. polycarpa* seed germination and the growth of sterile seedlings. However, excessive AC can adsorb PGRs, which may impair seedling growth. *Chen (2021)* reported that a low concentration of AC increased the germination rate of *Syzygium album* seeds, while higher concentrations had an inhibitory effect. The WPM medium has a lower concentration of inorganic salts, while MS is a rich, salt-balanced medium with higher concentrations of inorganic salts and ions. The 1/2 MS medium is a diluted version of MS, reducing trace elements by half. In the context of *I. polycarpa* seed germination and growth, 1/2 MS medium proved to be more effective than both MS and WPM, suggesting that reducing the concentration of inorganic salts in the medium enhances seed germination. This may be due to a more favorable osmotic environment and less stress on the seeds during early development.

### Effects of sucrose concentration and PGRs on root formation

Increasing the sucrose concentration in the culture medium has been shown to increase foliar starch and sugar contents while decreasing the water potential, leading to significant changes in root length, root number, and leaf development (*Badr, Angers & Desjardins, 2015*). *Gaspar et al. (2002)* found that the addition of sucrose to the culture medium hinders chlorophyll synthesis, disrupts the Calvin cycle and photosynthesis, and disturbs
overall carbon metabolism in plantlets cultured *in vitro*. In the present study, three different sucrose concentrations were tested for their effects on rooting in *I. polycarpa* seedlings. The best results were obtained at 10.0 g·L$^{-1}$, followed by 20.0 and 30.0 g·L$^{-1}$. However, in a study by *Hong et al. (2023)*, the optimal sucrose concentration for root formation by *I. polycarpa* was 20.0 g·L$^{-1}$. This discrepancy may be related to differences in the basic medium used, seed source, and/or environmental conditions.

The growth and differentiation of plant roots are closely related to the types and concentrations of PGRs. Previous studies have reported that NAA induces excellent rooting in coniferous tree species (*Browne, Davidson & Enns, 2000*; *Goldfarb et al., 2010*), whereas *Costa Junior et al. (2018)* found that IBA is more stable and thus more effective in inducing root formation in plant tissue culture. Similarly, in the present study, IBA was the strongest inducer of root formation in *I. polycarpa*. Differences in the rooting efficiency of tissue culture seedlings may also be related to the concentration of the PGR. Although *I. polycarpa* aseptic seedlings were induced to root in MS medium in the absence of PGRs, the roots tended to be sparse and underdeveloped and thus suboptimal for transplantation. However, the addition of an appropriate concentration of a single auxin significantly promoted rooting; following IBA supplementation at an optimal concentration of 0.7 mg·L$^{-1}$, rapid root differentiation resulted in a high number of healthy, robust roots. A higher IBA concentration inhibited rooting in *I. polycarpa* seedlings, increased callus formation at the plant base, and caused leaf yellowing. Additionally, the roots became shorter and thicker, indicating growth inhibition. This observation is consistent with the rapid IBA-induced propagation and rooting of *I. polycarpa* sterile seedlings reported by *Yang, Liu & Xie (2013)*, although the concentrations of IBA tested in that study were slightly different from those in ours. In a study by *Gianguzzi et al. (2024)*, the optimal concentration of IBA for root regeneration in *Hieracium lucidum* was 1.0 mg·L$^{-1}$, based on root length and number. IAA induces the production of fewer, longer, and more slender roots, generally without the formation of lateral roots. In a study of *Asarum sieboldii*, *Huang et al. (2016)* found that 0.6–0.8 mg·L$^{-1}$ NAA rapidly promoted root differentiation and growth, leading to the formation of robust roots. When the NAA concentration exceeded 0.05 mg·L$^{-1}$, shallow yellow-green callus tissue appeared at the base, and the roots became shorter, thicker, and enlarged, with a spongy consistency that made them vulnerable to breakage during seedling washing; transplanted seedlings had correspondingly higher death rates (*Li, 2023*). These findings are in contrast to those of our study, in which the optimal concentration of NAA was 0.05 mg·L$^{-1}$. This discrepancy may reflect differences in the basal culture medium and/ or sources of seeds and other experimental materials. Both IBA and NAA induced adventitious root formation and were more potent when used in combination than alone. *Pandey et al. (2019)* similarly showed that the rooting effect of auxins used in combination was better than that of a single agent. Furthermore, in practice, the rooting rate, root length, and root number are influenced not only by PGRs and their concentrations but also by factors such as seedling size, leaf quantity, and cultivation conditions. When cost is a concern, the use of 0.7 mg·L$^{-1}$ IBA alone is an effective and economical option for inducing rooting in *I. polycarpa* tissue culture seedlings.

### Effects of substrate ratios and transplant container types on tissue culture seedlings

During the transplantation process of *I. polycarpa* tissue culture seedlings, various factors can affect their survival rate and growth conditions. The substrate quality is a key factor for seedling development, making the selection of a suitable substrate essential for advancing the forestry and seedling industry. *Jiang et al. (2006)* found in their study on the transplantation of tissue culture seedlings of *I. polycarpa* that the optimal mixed substrate ratio was vermiculite (V): perlite (V): peat (V) = 6:3:1, with a survival time of 25 days and the survival rate reached 85%. Through comparative analysis of three formulations, the optimal substrate formulation was selected as follows: organic nutrient soil: perlite: vermiculite = 2:1:1 (V/V/V), with a survival rate of 96.67%. This formula has a long-lasting fertilizer effect, strong water and fertilizer retention ability, good permeability and drainage, robust seedling growth, and high quality. Therefore, the water and gas balance brought about by a reasonable ratio of transplantation substrate plays an important role in the growth of tissue culture seedlings of *I. polycarpa*.

Our investigation of the effects of conventional seedling trays *vs.* transparent plastic cups on the growth of *I. polycarpa* seedlings showed that seedling trays led to the formation of stunted, crooked roots. The impaired growth of these roots accounted for the low survival rate of the seedlings (73.33%). The survival rate of seedlings transplanted into transparent plastic cups was significantly higher and was accompanied by improved seedling height, ground diameter, and root elongation. The seedlings were first placed in a light chamber for 10 days with appropriate photoperiod stimulation and rapidly reached an autotrophic state that enhanced their resilience. The seedlings were then transferred to a greenhouse, where the controlled environment allowed their further stabilization. The transitional phase in the light chamber facilitated adjustment to the greenhouse, which promoted faster seedling growth and root development, thereby improving survival rates after transplantation. Because tissue culture seedlings differ significantly in their morphology and physiology from seedlings grown naturally from seeds, environmental conditions, substrate moisture, and transplantation timing were identified as critical factors affecting survival. Finally, using transparent plastic cups to transplant tissue culture seedlings directly into a mixed substrate eliminates the need for the traditional acclimation period in tissue culture bottles, significantly shortening the nursery cycle and reducing production costs.

In summary, we used *I. polycarpa* seeds as explants to establish an aseptic tissue culture-based rapid propagation system of this species. Because tissue culture seedlings differ significantly in their morphology and physiology from seedlings grown naturally, transplantation into a controlled environment such as a greenhouse is necessary to allow them to gradually adapt to natural conditions. In this study, we used transparent plastic cups to transplant tissue culture seedlings directly into a mixed substrate, eliminating the need for the traditional acclimation period in tissue culture bottles, significantly shortening the nursery cycle and reducing production costs.

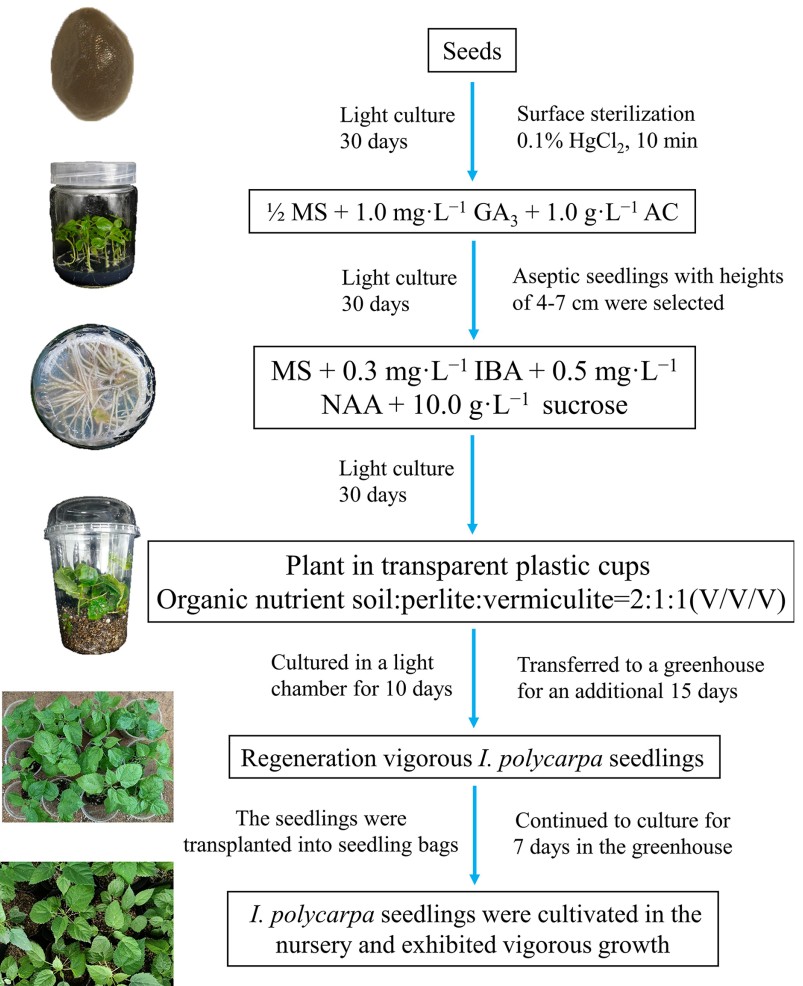

**Figure 3 Schematic model of the the rapid propagation system for *I. polycarpa*.**

This protocol enhances propagation efficiency by reducing the nursery cycle duration and production costs, offering an innovative framework for large-scale cultivation. However, further optimization is required in safety protocols, genetic uniformity of seed sources, and genetic variability of tissue-cultured plantlets. Practical implementation faces multiple challenges, including scalability barriers and a shortage of skilled personnel for extension services and training.

## CONCLUSIONS

In this study, we experimentally determined the optimal conditions for the germination and propagation of *I. polycarpa* seedlings. The best results in terms of germination and growth were obtained using disinfected seeds cultured on 1/2 MS + 1.0 mg·L$^{-1}$ GA$_3$ + 1.0 g·L$^{-1}$ AC + 30.0 g·L$^{-1}$ sucrose + 8.0 g·L$^{-1}$ agar. For rooting, the most suitable culture medium was MS + 0.3 mg·L$^{-1}$ IBA + 0.5 mg·L$^{-1}$ NAA + 10.0 g·L$^{-1}$ sucrose + 7.0 g·L$^{-1}$ agar. The most suitable conditions for transplanting *I. polycarpa* rooted seedlings involved direct

transplantation into transparent plastic cups in which the substrate was a mixture of organic nutrient soil, perlite, and vermiculite at a 2:1:1 volume ratio. Seedlings cultured in a light chamber for 10 days and then transferred to a greenhouse for an additional 15 days had a survival rate of 96.67%. This study established a rapid micropropagation protocol for *I. polycarpa* using seeds as explants. Figure 3 outlines the essential stages of the protocol and includes representative images of the process. Future research should focus on the differential regulation mechanism of IBA, NAA, and IAA on the rooting of tissue-cultured seedlings of *I. polycarpa*, and evaluate the differences in growth dynamics and physiological characteristics between *in vitro* regenerated seedlings and maternal plants. The optimized aseptic germination system developed in this study for *I. polycarpa* seeds improved seedling propagation efficiency and holds significant implications for the sustainable development of the *I. polycarpa* industry, advancing rural revitalization, and enhancing ecological conservation efforts.

### Funding

This work was supported by the Scientific Innovation Fund for Post-graduates of Central South University of Forestry and Technology Project (No. 2024CX02045), the Scientific Innovation Fund for Post-graduates of Hunan Project (No. CX20240694), the Science and Technology Innovation Plan Project of Hunan Province (No. 2022RC1155), the College Students' Innovation and Entrepreneurship Training Program Project of Hunan Province (No. S202410538003), the College Students' Innovation and Entrepreneurship Training Program Project "Establishment of *Idesia polycarpa* Tissue Cultivation and Regeneration System" of Hunan Province (No. 20200342) and the Shanghai Jintongxi Seed Industry Co., LTD. Key research and development project on *Idesia polycarpa* seed "Research on the selection and Breeding Technology of the superior seed of *Idesia polycarpa* Seed" (No. JKXM2022006). The funders had no role in study design, data collection and analysis, decision to publish, or preparation of the manuscript.

### Grant Disclosures

The following grant information was disclosed by the authors:
Post-graduates of Central South University of Forestry and Technology Project: 2024CX02045.
Scientific Innovation Fund for Post-graduates of Hunan Project: CX20240694.
Science and Technology Innovation Plan Project of Hunan Province: 2022RC1155.
Innovation and Entrepreneurship Training Program Project of Hunan Province: S202410538003 and 20200342.
Shanghai Jintongxi Seed Industry Co., LTD: JKXM2022006.

### Competing Interests

The authors declare that they have no competing interests.

## Author Contributions

- Zhangtai Niu conceived and designed the experiments, performed the experiments, analyzed the data, prepared figures and/or tables, authored or reviewed drafts of the article, and approved the final draft.
- Juan Xiao performed the experiments, analyzed the data, prepared figures and/or tables, and approved the final draft.
- Chuxi Hu performed the experiments, prepared figures and/or tables, and approved the final draft.
- Yunchen Yang performed the experiments, authored or reviewed drafts of the article, and approved the final draft.
- Sijing Shi performed the experiments, authored or reviewed drafts of the article, and approved the final draft.
- Xiaoyu Lu performed the experiments, prepared figures and/or tables, and approved the final draft.
- Yian Yin performed the experiments, prepared figures and/or tables, and approved the final draft.
- Ze Li conceived and designed the experiments, authored or reviewed drafts of the article, and approved the final draft.
- Lingli Wu conceived and designed the experiments, analyzed the data, prepared figures and/or tables, authored or reviewed drafts of the article, and approved the final draft.

## Data Availability

The raw measurements are available in the Supplemental File.

## Supplemental Information

Supplemental information for this article can be found online at http://dx.doi.org/10.7717/peerj.19395#supplemental-information.

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
