# Peer review of "Sterile seed germination and seedling cultivation of Idesia polycarpa"

_PeerJ, doi:10.7717/peerj.19395_

## Round 0.1 · original submission · Major Revisions

Please address the concerns of the two reviewers.

Reviewer 1 ·

Basic reporting

Please see the detailed comments.

Experimental design

Please see the detailed comments.

Validity of the findings

Please see the detailed comments.

Additional comments

Language issue
The current version of this manuscript may have been directly translated from another language using an online translation tool. This has led to issues with the fluency and coherence of the writing, significantly impacting the readability and overall quality of the manuscript. Furthermore, many sentences are overly long and convoluted, making the text difficult to follow. These structural issues need careful attention and revision to improve clarity and ensure that the scientific content is communicated effectively.
I strongly recommend that the authors address these major concerns by refining the language and restructuring the sentences where necessary. It would also be beneficial for the manuscript to be reviewed by a professional English editor or a native English speaker to ensure that the language is polished and the writing style meets the standards required for publication.
Enhancing the writing style and overall quality of the manuscript is crucial before it can be considered for potential publication. Therefore, I recommend to the Editor that this manuscript be returned to the authors for major revision. Once these improvements have been made, the manuscript can be re-evaluated for its suitability for publication.

Common correction
1. Use the full name of “Idesia polycarpa” rather than “I. polycarpa”.
2. Various writing issues are found, for example, misuse of the comma (,) and not adding the space after the comma. Some simple but essential corrections must be needed to be overcome.
3. Use the space before making a citation, for example, line 51 – “across China(Wang et al., 2023).”, line 55 – “diseases(Li et al., 2024)”, and so on (in whole manuscript).

Detail comments
Title
The manuscript title needed to be reconsidered

Abstract
The abstract needs to improve in terms of research background, methodology, results writing, and future implications.
Line 25-28: Rewrite the sentence and separate it into two.
Line 28-19: Remove the sentence.
Line 30-33: Rewrite,
Line 34-37: Rewrite the sentence and separate it into two.
Line 41-43: Correct the sentence (Grammar).
Line 43-45: It can be added to the methods section.
Line 40-46: Rewrite your results more clearly and use some significant findings in this section.

Introduction section
Line 56-87: Rewrite the sentence (focus on writing style and grammar), and read some related articles, especially on the same species.

Materials and methods section
Line 90-93: Rewrite and use the correct form of coordinates.
Line 95-131: Rewrite the sentence (focus on writing style and grammar), and read some related articles, especially on the same species.
Line 135-195: Rewrite the sentence (focus on writing style and grammar), and read some related articles, especially on the same species.
Line 197-198: This section should be “Statistical Analysis.” You need to add details about which method has been used to analyze the data, not only the software name (please read some recent articles).

Results section
Line 202-210: Rewrite and follow the research article writing style and improve the quality.
Line 212-225: need to improve this section.
Line 228-240: Rewrite the section and improvement required.
Line 243-245: Rewrite the sentence.
Line 249-252: Rewrite the sentence.
Line 257-277: Rewrite the section and make the required improvements.
Line 283-292: Rewrite the section and make the required improvements.
Line 296-330: This section must need to improve (writing style and grammar).

Discussion section
Line 332-433: The whole discussion section needs to improve by it’s improving the writing style, grammar, and formatting.
The author can consider using the subgroup in the discussion section.

Conclusions section
The conclusions seem to be a copy of the abstract results section. The conclusion section needs to be rewritten with precise results and future implications.

·

Basic reporting

The manuscript is well-structured, technically correct, and addresses a significant issue in the propagation of I. polycarpa. However, the articles lack a comprehensive and sufficient introduction and background which leaves a gap in demonstrating how their findings integrate with and contribute to the broader field of knowledge. The introduction provides a concrete background on the species and its significance, but limited detail on the previous studies that have attempted similar tissue culture techniques. If not, it is difficult to fully appreciate the significance and relevance of the work within the existing body of knowledge. You can find additional comments in the tracked changes within the attached PDF.
The structure of the articles, along with the organization of figures and tables, is professionally fulfilled and appropriately cited throughout the text, following the required standards, and is editable. Moreover, the availability of relevant raw data enhances the credibility of the research work, allowing transparency and facilitating potential replicability of further analysis (fulfilling the FAIR principles).
The research work is self-contained and comprises all necessary information for understanding the research without requiring the reader to refer to external sources. The article considerately includes all the results directly related to the hypotheses, making it easy to see how the findings connect to the research questions.
Comment on language and grammar issues: major modifications in terms of clarity of the idea, and grammar are important to improve the quality of the manuscript. Attention to grammatical detail and readability should be taken into account to improve the overall quality of the manuscript

Experimental design

Strength: The experimental design aligns with the original primary research focus within the aims and scope of the journal. The investigation is conducted carefully, following technical standards. The detailed methodology reflects cautious planning and performance, ensuring robust experimental procedures. The methods are described with sufficient detail to allow replication by other researchers. The procedures for seed preparation, sterilization, disinfection, and subsequent culture are strictly documented, including specific concentrations, and time frames.
The research question is well-defined, relevant, and meaningful, addressing a significant gap in the knowledge related to the germination, rooting, and transplanting techniques of I. polycarpa seeds.
Weakness: additional clarification on procedures, consistency in units, terminology, and formatting are needed to enhance the readability and simplicity. You can find the detailed comments on the change track. The rationale for selecting a specific treatment for example the concentration of each treatment should be explained more explicitly to strengthen the work. Some areas lack detailed information for complete reproducibility such as the rooting powder solution's exact formulation and the greenhouse conditions during the transplanting phase.

Validity of the findings

The study on the I. polycarpa seed germination and seedling development provides valuable understanding into improving tissue culture protocols for this species. The novelty of the work lies primarily in species-specific findings rather than in introducing groundbreaking methodologies. Thoroughly examining factors like disinfection times, culture media, activated carbon concentrations, plant growth regulators, rooting agents, and transplanting conditions, the research presents practical guidelines to enhance the efficiency of I. polycarpa tissue culture.
The primary contribution is the adaptation and refinement of existing tissue culture techniques to the unique needs of I. polycarpa. However, the approaches are well-established in the field, and the specific responses and optimal conditions identified for I. polycarpa add to the species-specific knowledge base. This is particularly important for anyone looking to cultivate this plant under controlled conditions for conservation purposes.
In summary, the work is novel in its focus on application to I. polycarpa, providing an inclusive protocol tailored to this species. While it may not introduce entirely new concepts to plant tissue culture, it significantly advances our understanding and capability to cultivate I. polycarpa effectively.

Additional comments

The manuscript addressed a critical research gap in Sterile seedlings cultivation and transplanting Idesia polycarpa seeds. The topic raised in this manuscript holds significance for forest managers. However, a limitation of the study is the lack of detailed information in the introduction part related to the existing studies on tissue culture work, grammar, and clarity of methods. Thus, the focus should be given into account to improve the manuscript and easily understandable by the readers.
There are grammatical errors that need to be addressed as well and that diminish understanding of the manuscript. I suggested a major language edition to make the points clearer, concise, and easier to read and understand.
The introduction sections still require significant refinement to enhance conciseness and clearly articulate the study's objectives. Thus, I encourage the Authors to pay more attention to improving the content of the introduction, making it more engaging for readers.
The methodology section lacks some clarity that is essential for comprehending the results and should be incorporated.
In my opinion, you must be more specific about what is the take-home message that you want to communicate in your conclusion. Thus, I recommend a minor revision of the conclusion part.

---

## Round 0.2 · Major Revisions

Reviewer 1 has provided extensive additional comments in their appended PDF. The line numbers they refer to correspond to the line numbers of your 'clean' Word document. There are extensive comments related to readability and language.

Reviewer 1 ·

Basic reporting

Please see the Review report and attachment.

Experimental design

Please see the Review report and attachment.

Validity of the findings

Please see the Review report and attachment.

Additional comments

Please see the Review report and attachment.

Annotated reviews are not available for download in order to protect the identity of reviewers who chose to remain anonymous.

·

Basic reporting

Dear Editor,

I apologize for the delay in completing my review because I was engaged with another task.

I have carefully reviewed the manuscript entitled "Sterile Seed Germination and Seedling Cultivation of Idesia polycarpa," along with the author's response to my previous comments. I am pleased to report that the authors have thoroughly addressed the points I raised during the initial review process. The manuscript now reflects significant improvements, particularly in grammar and clarity, as well as a strengthened background section that enhances the context and relevance of the study.

Overall, the manuscript is well-organized and adheres to the structure of a professional article, presenting the information clearly and logically. Based on these improvements, I believe the manuscript has the potential to make a meaningful contribution to the field.

Experimental design

Dear Editor,

I have reviewed the revised manuscript entitled "Sterile Seed Germination and Seedling Cultivation of Idesia polycarpa," with a particular focus on the experimental design, which I commented on in my previous review. I am pleased to note that the authors have made significant improvements in this area, and I am satisfied with the changes they implemented.

The authors have responded to my questions and comments thoroughly and constructively, adhering to technical and ethical standards expected within the scientific community. They have effectively addressed the previous lack of clarity, ensuring consistency in units, terminology, and formatting. These adjustments have notably enhanced the readability and accessibility of the manuscript.

In its current form, I find the manuscript to be clearer and more informative, providing valuable insights for readers. I appreciate the authors' efforts in making these revisions and believe the manuscript has substantially benefited from them.

Validity of the findings

Dear Editor,

In my previous comments, I highlighted that the novelty of this manuscript lies predominantly in the species-specific insights it provides, rather than in the introduction of groundbreaking methodologies. While the study does not propose entirely new techniques within the field of plant tissue culture, it nonetheless presents valuable findings specific to Idesia polycarpa, offering a comprehensive protocol tailored to the unique requirements of this species.

The work makes a meaningful contribution by expanding our understanding of the optimal conditions for sterile seed germination and seedling cultivation for I. polycarpa. This contribution is particularly valuable as it addresses practical challenges associated with cultivating this species, thereby advancing current knowledge and enhancing cultivation efforts. Such advancements may serve as a foundation for future studies and facilitate wider applications for species-specific protocols in plant tissue culture.

Moreover, I have noted an improvement in the manuscript's conclusion, which is now clearer and more comprehensive compared to the previous version. The revised conclusion effectively summarizes the study's significance and practical applications, reinforcing its value for researchers and practitioners in the field.

In summary, while the work may not push the boundaries of methodological innovation, its species-centered approach provides an important addition to the literature, with direct implications for those working with I. polycarpa and similar species.

Additional comments

Dear Editor,

Overall, I have positive feedback on the revised manuscript entitled "Sterile Seed Germination and Seedling Cultivation of Idesia polycarpa." The authors have addressed previous feedback comprehensively, resulting in a well-structured and clear manuscript that holds significant value for both the scientific and practical understanding of this species.

This study provides insights into species-specific cultivation methods for I. polycarpa, offering practical protocols that can be applied to improve the species’ propagation and conservation. The findings, though focused on one species, have broader implications, as they contribute to the body of knowledge on sterile seed germination and seedling development. This work will be particularly beneficial to the local community where the species is native, supporting conservation and potential economic use. Additionally, it presents a relevant contribution to the global scientific community by advancing understanding of species-specific propagation techniques, which could be adapted to similar species.

Based on these strengths, I am confident that this manuscript represents an important and constructive addition to the field. I recommend it for publication and believe it will be well-received by researchers and practitioners alike.

Thank you for considering my recommendation. Please let me know if any further clarification is needed.

Best regards,
Tamiru

---

## Round 0.3 · Major Revisions

Dear authors, your manuscript needs serious revision and improvement of the language quality. If next time the language quality does not meet international publishing standards, I will be forced to refuse your publication.

Reviewer 1 ·

Basic reporting

Based on a thorough assessment of the manuscript, it is unsuitable for publication in its current form. I have highlighted sections throughout the manuscript where improvements are necessary. The highlighted sections in yellow indicate grammatical issues, and the line numbers below indicate areas where the writing lacks clarity or conciseness.

Experimental design

Please see the attachment

Validity of the findings

Please see the attachment

Additional comments

Please see the attachment

Annotated reviews are not available for download in order to protect the identity of reviewers who chose to remain anonymous.

---

## Round 0.4 · Major Revisions

Dear authors, I ask you to carefully correct each of the reviewers' fundamental comments and hope that the new version of this manuscript can be approved by them for publication.

Reviewer 3 ·

Basic reporting

• Clarity and Language:
The manuscript is well-written in formal academic English. However, certain sentences are quite dense, which may affect readability. A more concise and structured approach would enhance clarity. Minor grammatical corrections may also improve the flow.
• Literature Review and Context:
The introduction provides a thorough background on Idesia polycarpa, its significance, and the need for improved germination techniques. The literature is well-referenced, though incorporating more recent studies on tissue culture techniques could strengthen the review.
• Manuscript Organization and Formatting:
The manuscript adheres to PeerJ standards. However, subsections within the methodology and results could be better delineated for improved readability.
o Figures and Tables: The provided figures and tables are appropriate, but clearer labeling and references in the text would enhance comprehension.
o Raw Data Availability: The raw data is provided, but it should be explicitly referenced in the manuscript to ensure transparency.

Experimental design

• Research Scope and Objectives:
The study is well within the journal’s scope, addressing an important agricultural and ecological challenge. The research question is well-defined and relevant.
• Methodological Rigor:
o The disinfection process and germination techniques are detailed but could benefit from a clearer step-by-step breakdown.
o While the selection of plant growth regulators (PGRs) is justified, the reasoning behind the specific concentrations chosen is not always clear. More context from previous studies could enhance this section.
o The statistical methods are appropriate but require more explicit justification—why were specific tests chosen, and how do they strengthen the study’s conclusions?
• Reproducibility and Transparency:
o The methodology is sufficiently detailed for replication, though flowcharts or diagrams summarizing key experimental steps could improve clarity.
o If additional datasets were used, a data availability statement should be included.

Validity of the findings

• Scientific Soundness:
The results align with the research question and are supported by data. The authors successfully demonstrate the efficacy of various germination and propagation techniques for Idesia polycarpa.
• Key Strengths of the Study:
o The study presents a comprehensive approach to improving germination rates.
o The plant tissue culture methodology is well-structured and provides useful insights into sterile seedling cultivation.
• Areas for Improvement:
o The limitations of the study are not thoroughly discussed. It would be beneficial to include a section on potential biases or challenges in real-world application.
o The discussion should more explicitly compare findings to previous research, highlighting where results align or diverge.
o The conclusion would benefit from more specific practical applications—how can farmers or researchers use these findings?

Additional comments

• The study contributes valuable knowledge on the propagation of Idesia polycarpa and presents promising techniques for improving germination.
• The manuscript would benefit from structural refinements, particularly in the results and discussion sections, to enhance readability and logical progression.
• The figures and tables should be more clearly referenced in the text to facilitate understanding.
• Future research directions are briefly mentioned but could be expanded with more concrete recommendations.
* * *
Final Recommendation
The manuscript is scientifically sound and highly relevant, but it requires minor revisions to:
1. Improve clarity and conciseness.
2. Strengthen methodological transparency.
3. Enhance the discussion of results in relation to existing literature.
Once these refinements are addressed, the study will be a strong candidate for publication in PeerJ.

·

Basic reporting

An interesting article with extensive research and precise presentation of the material.
At the beginning of the article, the authors point out the problem of seed dormancy. In discussing the results, it would be desirable to provide a more detailed description of the mechanisms of seed dormancy formation and overcoming.
Keywords: Idesia polycarpa; tissue culture; aseptic germination; acclimatization and transplanting; dormancy control.
Materials & Methods in the methods do not indicate the purity level of the reagents used, it is advisable to indicate the manufacturer.
Line 277 Rooting culture of I. polycarpa aseptic seedlings.
Hereafter: Aseptic growing conditions, but sterile seedlings, solution, etc.
Figure 1 is of high quality. Tables 8 and 9 discuss the color and shape of the plants; at the discretion of the authors, photographs of the results are desirable.
Please check the Row_Data in Table 5: Root Length (cm) does not match Average Root Length (cm).

Experimental design

No comment

Validity of the findings

No comment

---

## Round 0.5 · accepted · Accept

Dear Dr. Niu, Thank you for a good article. I hope it will generate considerable interest among readers and will receive a large number of citations.

Reviewer 3 ·

Basic reporting

The authors have considered all the recommendations of the reviewer. The manuscript has been significantly improved after the revision.

Experimental design

The authors have considered all the recommendations of the reviewer. The manuscript has been significantly improved after the revision.

Validity of the findings

The authors have considered all the recommendations of the reviewer. The manuscript has been significantly improved after the revision.

Additional comments

I recommend the article to be published

·

Basic reporting

no comment

Experimental design

no comment

Validity of the findings

no comment

Additional comments

Line 35: activated carbon (AC) - the abbreviation is not supported in the text